# Follistatin Is a Novel Chemoattractant for Migration and Invasion of Placental Trophoblasts of Mice

**DOI:** 10.3390/cells11233816

**Published:** 2022-11-29

**Authors:** Jing Li, Yan Qi, Ke Yang, Linjing Zhu, Xueling Cui, Zhonghui Liu

**Affiliations:** 1Department of Immunology, College of Basic Medical Sciences, Jilin University, Changchun 130021, China; 2Key Laboratory of Neuroimmunology and Clinical Immunology, Changchun 130021, China; 3Institute of Applied Technology, Hefei Institutes of Physical Science, Chinese Academy of Sciences, Hefei 230031, China; 4Department of Genetics, College of Basic Medical Sciences, Jilin University, Changchun 130021, China

**Keywords:** trophoblasts, migration, follistatin, JNK signaling

## Abstract

Follistatin (FST) as a gonadal protein is central to the establishment and maintenance of pregnancy. Trophoblasts’ migration and invasion into the endometrium are critical events in placental development. This study aimed to elucidate the role of FST in the migration and invasion of placental trophoblasts of mice. We found that FST increased the vitality and proliferation of primary cultured trophoblasts of embryonic day 8.5 (E8.5) mice and promoted wound healing of trophoblasts. Moreover, FST significantly induced migration of trophoblasts in a microfluidic device and increased the number of invasive trophoblasts by Matrigel-coated transwell invasion assay. Being treated with FST, the adhesion of trophoblasts was inhibited, but intracellular calcium flux of trophoblasts was increased. Western blotting results showed that FST had no significant effects on the level of p-Smad3 or the ratio of p-Smad3/Smad3 in trophoblasts. Interestingly, FST elevated the level of p-JNK; the ratio of p-JNK/JNK; and expression of migration-related proteins N-cadherin, vimentin, ezrin and MMP2 in trophoblasts. Additionally, the migration of trophoblasts and expression of N-cadherin, vimentin, and MMP2 in trophoblasts induced by FST were attenuated by JNK inhibitor AS601245. These findings suggest that the elevated FST in pregnancy may act as a chemokine to induce trophoblast migration and invasion through the enhanced JNK signaling to maintain trophoblast function and promote placental development.

## 1. Introduction

Placental trophoblasts are major building blocks of placenta, which have capacity of invading deeply into the decidualized maternal uterine tissue to remodel the uterine spiral arteries so as to favor embryo development. These abilities are crucial in ensuring the formation of a nutrient-supplying functional placenta. Defects in placental function can result in fetal growth retardation in the second or third trimester of pregnancy in humans and in the second half of gestation in mice such as pre-eclampsia, miscarriage, and ectopic pregnancy [1,2]. Additionally, placental trophoblasts provide a main source of pregnancy-associated hormones and cytokines to protect the semi-allogeneic fetus from immune rejection, such as EGF, IL-1β, and TNF-α et al. [3]. Thus, we have to take notice of cell molecular mechanisms in the early steps of placental development to better understand the most common pregnancy complications.

Follistatin (FST) can suppress follicle stimulating hormone (FSH) release of the pituitary and act as a negative regulator by binding activin with high affinity, so as to prevent activin from binding to its receptors [4,5,6]. Previous studies have reported that activin A and FST levels in serum rise progressively throughout pregnancy [7]. In humans, aberrant expression of FST and activins has led to serious consequences in women, such as recurrent miscarriage [8,9], hypertensive disorders during pregnancy [10], and repeated implantation failure after in vitro fertilization [11]. Furthermore, absence of FST in mice has led to poor capacity of uterine receptivity, with lower levels of proliferation and differentiation in decidualized uterus tissue [12]. The knockout and aberrant overexpression of FST frequently cause perinatal mortality in mice [13] and female infertility resulting in thin uteri [14]. However, the role of FST in migration and invasion of placental trophoblasts remains unclear.

Epithelial to mesenchymal transition is the process characterized by the loss of cell-cell adhesion, reorganization of cytoskeleton and expression of matrix metalloproteinases (MMPs), all of which are present in invasion of trophoblasts, metastasis of cancer cells, and gastrulation [15,16]. In humans and mice, subsequent to the blastocyst adhesion to the uterine wall, the trophoblasts invade into the endometrial matrix with a switch from a polarized epithelial phenotype to a highly motile mesenchymal phenotype [17]. The studies have reported that transforming growth factor beta (TGF-β) has a predominant role in the induction and progression of EMT through Smads signaling and non-Smads signaling, such as p38, ERK1/2, and JNK. 

In the present study, the primary cultured trophoblasts of mice in the early stage of pregnancy were utilized to investigate the effects of FST on migration and invasion of placental trophoblasts. We found that FST promoted vitalities of trophoblasts and induced migration and invasion of trophoblasts through JNK signaling, proving for the first time that FST is a novel chemoattractant for the migration and invasion of trophoblasts to be beneficial for placental development.

## 2. Materials and Methods

### 2.1. Reagents

Recombinant human FST288 was purchased from R&D systems (Minneapolis, MN, USA). Cell Counting Kit-8 (CCK-8) was bought from GlpBio Biotechnology Co. (Shanghai, China). Rabbit anti-GAPDH polyclonal antibody and JNK inhibitor AS601245 were provided by Absin (Shanghai, China). Rabbit anti-vimentin, anti-N-cadherin, and anti-ezrin polyclonal antibodies were obtained from Cell Signaling Technology (Danvers, MA, USA). Rabbit anti-E-cadherin polyclonal antibody was purchased from Proteintech (Chicago, IL, USA). Rabbit anti-JNK and anti-p-JNK antibodies were provided by Wanlei (Shenyang, China). Rabbit anti-MMP2, anti-p-Smad3 and anti-Smad3 polyclonal antibodies were provided by Abclonal (Woburn, MA, USA). Fluo-4 AM was purchased from Thermo Fisher Scientific (Ottawa, ON, Canada). Giemsa staining and FITC-labeled goat anti-rabbit IgG were provided by Sigma-Aldrich (Oakville, ON, Canada).

### 2.2. Cell Culture

All mice experimental protocols were performed as approved by the Animal Committee of Jilin University (No 2021-214). Male and female BALB/c mice of 8–10 weeks of age were mated in the same cage, and then the next morning, the vaginal plugs of female mice were checked. The day when the vaginal plugs were found was considered as embryonic day 0.5 (E0.5). The placentas were collected from female mice at embryonic day 8.5 (E8.5). Fetus, fetal membranes, and additional decidual tissue were removed from the surface of the placenta by gently peeling with forceps on ice. Then, placental tissue was quickly minced with scissors or surgical blades and transferred into a phosphate buffer solution (PBS) supplemented with 2% fetal bovine serum (FBS) (2% FBS-PBS). Chopped tissues were digested with serum-free medium containing 2 mg/mL DNase and 0.1 mg/mL liberase at 37 °C for 40 min. The undigested tissues were separated using a cell strainer, and the obtained suspension containing placental trophoblasts were centrifuged 300× *g* for 5 min. Placental trophoblasts were cultured in 10%-FBS-DMEM-F12 at 37 °C in a humidified incubator containing 5% CO_2_. At E8.5, more than 40,000 living trophoblast cells were obtained from a mouse placenta, and at least 95% of the living cells expressed CK-7 (Appendix A) [18,19,20].

### 2.3. Pretreatment of Primary Cultured Trophoblasts of Mouse with AS601245

Primary cultured placental trophoblasts of mice were pretreated with vehicle control [dimethylsulfoxide (DMSO)] (Ctrl) or 1 μmol/L JNK inhibitor AS601245 in DMSO for 1 h before treatment with 0 or 10 ng/mL FST. Next, Western blotting, microfluidic migration assay and Matrigel-coated transwell invasion assay were conducted according to the methods below.

### 2.4. CCK-8 Assay

Cell viability was determined using CCK-8 assay. The trophoblasts of mice (1 × 10^4^ cells per well) were seeded into a 96-well plate in triplicate and incubated in 2% FBS-DMEM-F12 containing 0–20 ng/mL FST at 5% CO_2_ for 24 h and 48 h, respectively. Next, 10 μL of CCK-8 reagent was added to the culture medium (100 μL) of each well, and then cells were incubated for 4 h at 37 °C. The absorbance was detected at 450 nm and 650 nm using a microplate spectrophotometer (Gene Company Limited, Shanghai, China). 

### 2.5. Real-Time Cell Analysis

Real-time cell analysis (RTCA) is a technology based on the principle of the microelectronic biosensor chip, which can realize the real-time analysis of cells without markers in the process of experiments [21]. The RTCA instrument (xCELLigence RTCA S16; ACEA Biosciences, SD, CA, USA) was used to analyze the proliferation and adhesion properties of primary cultured trophoblasts of mice. Briefly, 50 μL of culture medium was added to the well of an E16 xCELLigence microtiter plate, which was then inserted into the RTCA device. After 1 min, the background impedance was measured for each well. Subsequently, trophoblasts of mice (1 × 10^4^ cells in 50 μL culture medium) were added to each well and cultured for 2 h and then treated with different concentrations of FST for another 72 h in the proliferation assay. In the adhesion assay, trophoblasts of mice and FST were added to each well together and cultured for 5 h. Cells were all monitored in 15 min intervals. The cell index (CI) reflected the biological status of monitored cells, including the cell number, viability, morphology and adhesion. 

### 2.6. Giemsa Staining

The trophoblasts of mice were cultured in 2% FBS-DMEM-F12 without or with 10 ng/mL FST in a 96-well plate. After 24 h, cells were fixed with 4% paraformaldehyde, stained with Giemsa solution and observed under an optical microscope.

### 2.7. Wound Healing Assay

The trophoblasts (3 × 10^5^/well) of mice were seeded on a 12-well plate and allowed to reach 90% confluency overnight. Then, a scratch-wound was made in the monolayer cells using a 10 μL pipette tip, and the debris and floating cells were removed by washing twice with PBS. Cells were cultured in 2% FBS-DMEM-F12 containing 0–20 ng/mL FST. For each group, the images were taken at 0, 12, and 24 h after wounding, respectively, and the distance from wound side to side was measured. Independent experiments were repeated three times.

### 2.8. Microfluidic Cell Migration Assay

#### 2.8.1. Microfluidic Chip Preparation

The standard photolithography and soft-lithography techniques were used for fabricating the microfluidic device with quadruple testing units (D^4^-Chip). The prototype of the D^4^-Chip was designed using the Solidworks software (2018 version, Dassault Systems, MA, USA) and then manufactured in the Micro/Nano Research and Manufacturing Center, University of Science and Technology of China [22]. The geometry of the microchannel was defined by patterning two layers of SU-8 photoresist (MicroChem Corporation, Westborough, MA, USA) with different thicknesses on a silicon wafer through double exposures. Polydimethylsiloxane (PDMS) (Sylgard 184, Dow Corning) was poured on the silicon wafer to create the main body of the D^4^-Chip by soft-lithography technology. The PDMS replica with the inlet and outlet holes punched was then bonded to a commercial glass slide (Official of Jiangsu Shitai Experimental Equipment Co., LTD, 75 mm × 51 mm × 1.2 mm) by air plasma treatment (PDC-002, Micono Technology Co., LTD). Then, D^4^-Chip channels were then coated with 0.4% BSA for 60 min at 37 °C before cell migration experiments. The detailed structures of the D^4^-Chip are shown in Figure 1A–D.

#### 2.8.2. Cell Loading

Before loading the trophoblasts of mice, all of the holes of the D^4^-Chip were emptied. The 10 μL cell suspension (1 × 10^4^ cells per well) were then loaded from the four-cell loading inlets. Due to the pressure difference between the cell loading inlets and outlets, trophoblasts were pushed and aligned beside the cell localization channels (Figure 1E).

#### 2.8.3. Gradient Generation

Four gradient conditions were generated in the D^4^-Chip, including the medium-medium, medium-FST (5 ng/mL), medium-FST (10 ng/mL), and medium-FST (20 ng/mL) (Figure 1E). Equal volume of chemoattractant solution and medium were added into corresponding inlets at the same time. Thus, a stable chemical concentration gradient was formed in the main channel for cell migration. To maintain the gradient, the waste in the outlet was discarded and the appropriate equal volume of chemoattractant solution and medium was supplemented.

#### 2.8.4. Image Capture

The migration progress of the trophoblasts was recorded as images by using an inverted microscope camera. The image capture frequency was set as 1 frame per 2 h for 12 h to track cell trajectories. The microfluidic device was kept in a 37 °C incubator containing 5% CO_2_ between the imaging time points.

#### 2.8.5. Image Analysis

The average cell migration distances were measured and analyzed by using the Image J software. All migrated trophoblasts were chosen and tracked in the same field. The trophoblast chemotaxis was expressed by the average migration distance of migrating cells in the same size fields of microfluidic device in three separate experiments. 

### 2.9. Matrigel-Coated Transwell Invasion Assay

The invasion abilities of trophoblasts of mice were evaluated by Matrigel-coated transwell invasion assay (8 μm pore size; Corning, CO, NY, USA) [23,24,25]. Briefly, the inserts of transwell chamber were coated with 200 μg/mL Matrigel (Corning, CO, NY, USA) that was diluted with 100 μL ice-cold serum-free medium. The trophoblasts of mice (5 × 10^4^ cells in 200 μL DMEM-F12 containing 2% FBS) were seeded into the upper chamber, and 750 μL culture medium containing 0, 5, 10, 20 ng/mL of FST was added into the lower chambers, respectively. After incubation for 10 h, the non-migrated cells on the upper side of the chamber membranes were removed with cotton swabs. Cells on the bottom side of the chamber membrane were fixed with 4% paraformaldehyde for 20 min, stained with Giemsa and photographed. Cell numbers were counted in five randomly chosen fields from each chamber. Experiments were repeated three times.

### 2.10. Immunofluorescence Assay

In D^4^-Chips, the trophoblasts of mice (1 × 10^4^ cells per well) were loaded to the parallel test units of the microfluidic device from the cell inlets and aligned in the docking structures. An equal volume of chemoattractant FST solution and culture medium were added to the pairs of dosing inlets to configure different gradient conditions. After 4 h, PBS was added into the outlets and cell inlets of the chip for washing twice. Then, 4% paraformaldehyde was used to fix the cells for 30 min. The cells were permeabilized with 0.3% Triton-100 for 20 min, and then cells were washed three times with PBS for 5 min each time. Furthermore, the cells were blocked in goat serum for 30 min at 37 °C, and incubated with rabbit anti-ezrin antibody (1:200) at 4 °C overnight. The cells were washed three times with PBS for 5 min each time, and incubated with FITC-labelled goat anti-rabbit IgG antibody (1:100 dilution) at 37 °C for 30 min. Finally, nuclear counterstaining was conducted with DAPI for 10 min. Fluorescence images were recorded using the Zeiss 20X (0.8NA) Plan-Apochromat objective lens under an inverted fluorescence microscope.

### 2.11. Calcium Flux Assay

The trophoblasts of mice were incubated in 2% FBS-DMEM-F12 medium with 4 μmol/L Fluo-4 in dark for 40 min at 37 °C. Cells were washed twice using the 2% FBS-DMEM-F12 and transferred to flow cytometric tubes. The baseline signal data (F^0^) was recorded for 1 min. Then, Fluo-4 signals of cells (F) stimulated with 0 or 10 ng/mL FST were recorded instantaneously for another 4 min. FlowJo software (FlowJo_v10.6.2, FlowJo LLC., Ashland, OR, USA) was used to analyze the kinetics of Fluo-4 fluorescence intensity. The Fluo-4 fluorescence intensity was normalized to the baseline for comparison (F/F^0^). Independent experiments were repeated three times. 

### 2.12. Western Blotting

The trophoblasts of mice were lysed using a protein extraction reagent (Thermo Fisher Scientific, USA) supplemented with the protease and phosphatase inhibitor cocktail (Thermo Scientific, USA) and 0.5 mol/L EDTA solution. Proteins were quantified using a Pierce BCA protein assay kit (Thermo Scientific, USA) following the manufacturer’s instructions. Proteins (20 μg) were separated by 8% or 10% sodium dodecyl sulfate-polyacrylamide gel electrophoresis (SDS-PAGE) and transferred to polyvinylidene difluoride membranes (Millipore, USA). Membranes were blocked in 5% skimmed milk in TBS-T at room temperature for 2 h with gentle shaking and then incubated with the primary antibodies overnight at 4 °C. Membranes were further incubated with secondary antibody conjugated with horseradish peroxidase for 2 h at room temperature and followed by ECL detection (GE Healthcare, UK). Finally, membranes were scanned with a LAS-4000 luminescent image analyzer (Fujifilm, Japan). Independent experiments were repeated three times. 

### 2.13. Statistical Analysis

All data were presented as mean ± standard deviation (SD). Statistical analysis was performed using Student’s *t*-test or a one-way ANOVA followed by Tukey’s multiple comparisons test. The difference at *p* < 0.05 was considered statistically significant.

## 3. Results

### 3.1. FST Increased Viability and Proliferation of Primary Cultured Trophoblasts of Mouse

The viability and proliferation of trophoblasts are associated with trophoblast movement and placental function during pregnancy. Firstly, E8.5 mice primary cultured trophoblasts were stained by the immunofluorescence staining with anti-cytokeratin-7 (CK7) and Vimentin antibodies, in which at least 95% of the cells positively expressed cytokeratin-7 as a trophoblast marker, while Vimentin as a marker of mesenchymal cells was expressed negatively (Appendix A) [18,19,20]. Moreover, the viabilities of trophoblasts of mice treated with FST were examined by the CCK-8 assay. As shown in Figure 2A, compared with the culture medium control, 5, 10, and 20 ng/mL of FST increased the viabilities of primary cultured trophoblasts of mice for 24 h and 48 h, respectively. In addition, RTCA results also showed FST significantly promoted the proliferation of primary cultured trophoblasts of mice (Figure 2B).

### 3.2. FST Promoted Wound Healing of Primary Cultured Trophoblasts of Mouse

To further analyze effects of FST on trophoblast activity, morphology, and wound healing of primary cultured trophoblasts of mice were determined after treatment with FST for 24 h. We found that the morphology of trophoblasts treated with 10 ng/mL of FST exhibited a longer and polygonal shape, compared with that in control (Figure 3A). Moreover, a scratch wound healing assay showed that FST promoted wound healing of primary cultured trophoblast cells of mice in a dose-dependent manner (Figure 3B,C). Additionally, the migratory cells in the wound healing assay were also stained with cytokeratin-7 (CK7) (Appendix A), which confirmed that the migratory cells in the wound healing assay are indeed trophoblasts that expressed CK7 positively (Appendix A). EMT is a critical process for embryogenesis, organogenesis, and cancer development [26,27]. In the process of placenta formation, the migration and EMT process of trophoblasts occurred [28]. Our data suggested that FST-treated trophoblasts might undergo EMT and cell migration.

### 3.3. FST Induced Migration and Invasion of Primary Cultured Trophoblasts of Mice

In this study, the D^4^-Chip of a microfluidic device was used to assess the chemotactic migration of trophoblasts towards FST gradients. The results revealed that FST induced the migration of primary cultured trophoblasts of mice in a time-dependent and dose-dependent manner. Compared with the culture medium control group, FST not only increased more migrated cells, but also enhanced greater distance of cell migration (Figure 4A–C).

The Matrigel-coated transwell invasion assay was further conducted to evaluate the effects of FST on the invasion of trophoblasts of mice. As shown in Figure 4D, compared with the culture medium control group, FST significantly induced the increased invasion of primary cultured trophoblasts of mice in a dose-dependent manner. The above data indicated that the increased FST as a chemokine in pregnancy might be involved in inducing migration and invasion of trophoblasts into the uterine wall. 

### 3.4. FST Suppressed Adhesion of Primary Cultured Trophoblasts of Mouse

Generally, the polarized epithelial cells lose their adhesion property and obtain mesenchymal cell phenotypes in the process of EMT in embryogenesis and diseases [29]. Therefore, RTCA was conducted to examine the effect of FST on the adhesion of trophoblasts of mice. The results showed that 10 ng/mL and 20 ng/mL of FST significantly suppressed the adhesion of primary cultured trophoblasts of mice (Figure 5). Hence, it is a possibility that FST induce trophoblast migration because it inhibits trophoblasts adhesion.

### 3.5. FST Increased Expression of Migration and Polarity-Related Protein of Trophoblasts

To discuss the potential mechanism of FST on migration and invasion of trophoblasts of mice, the expression of migration-related proteins was examined by Western blotting. The results showed that FST increased the protein expression of N-cadherin, vimentin, and MMP2 (Figure 6A), which were involved in EMT as well as cell migration and invasion.

Ezrin has the function of connecting the plasma membrane and the actin cytoskeleton and is involved in cell adhesion, membrane edge fluctuation and microvilli formation [30]. Additionally, ezrin is a polarity marker that plays an important role in cell polarity establishment, migration, and division [31]. Because FST induced the migration of mice trophoblasts, we further examined the effect of FST on the expression and localization of ezrin in trophoblasts. Western blotting results showed that FST increased the protein level of ezrin in primary cultured trophoblasts of mice (Figure 6B). Moreover, in the absence of FST gradient, the ezrin staining had any expression and distributed randomly in the cytoplasm and periphery of non-polarized trophoblast cells. However, in the presence of FST gradient, the ezrin expressed highly compared with the culture medium control group and concentrated at the direction of FST gradient in polarized trophoblasts (Figure 6C). These results suggested that FST-induced cell polarization might underlie effect on the chemotactic migration of trophoblasts. 

### 3.6. FST Increased Intracellular Calcium Flux and p-JNK Protein Level of Primary Cultured Trophoblasts of Mice

Previous studies have shown that cell migration is closely related to intracellular calcium flux [32]. In this study, we measured the intracellular calcium level in trophoblasts by flow cytometry. The results indicated that FST elevated the intracellular calcium levels in primary cultured trophoblasts of mice (Figure 7A). The effect of FST on intracellular calcium levels is consistent with the effect on inducing migration of trophoblasts.

To further explain how FST influences the migration and invasion of trophoblasts of mice, levels of Smad3 protein as activin canonical signaling molecule and JNK protein as MAPK signaling molecule were examined by Western blotting. We found that levels of p-Smad3 and p-Smad3/Smad3 did not change significantly in trophoblasts treated with 10 ng/mL of FST (Figure 7B). Interestingly, the levels of p-JNK and ratio of p-JNK/JNK were significantly increased (Figure 7C). We hypothesized that the enhanced JNK signaling might be involved in migration and invasion of primary cultured trophoblasts of mice.

### 3.7. JNK Inhibitor AS601245 Suppressed FST-induced Migration and Invasion of Primary Cultured Trophoblasts of Mice

AS601245, a c-Jun NH_2_-terminal kinase (JNK) inhibitor, indiscriminately inhibit phosphorylation of JNK substrates. In order to determine that JNK signaling was involved in regulating the migration and invasion of mice trophoblasts induced by FST, we examined migration and invasion of trophoblasts pre-treated with the JNK inhibitor AS601245. The results revealed that AS6011245 treatment attenuated the migration of trophoblasts induced by 10 ng/mL FST (Figure 8A,B). The same result was obtained with the Matrigel-coated transwell invasion assay (Figure 8C,D). Western blotting results showed that pretreatment with AS601245 decreased the p-JNK levels and the ratio of p-JNK/JNK in primary cultured trophoblasts treated with 10 ng/mL FST (Figure 8E). Concomitantly, AS601245 also attenuated the protein levels of MMP2, N-cadherin, and vimentin in primary cultured trophoblasts treated with 10 ng/mL FST (Figure 8E). These data further indicated that FST promoted the expression of migration-related proteins to facilitate cell migration and invasion through JNK signaling.

## 4. Discussion

FST that elevates progressively during pregnancy plays an important role in the establishment and maintenance of normal pregnancy. The migration and invasion of trophoblasts contribute to the remodeling of uterine spiral arteries and maintaining of a nutrient-supplying functional placenta. In the present study, our data demonstrated that FST induced migration of primary cultured trophoblasts of mice significantly in vitro with a microfluidic device and invasion by Matrigel-coated transwell invasion assay, which were related to increasing intracellular calcium flux and enhancing JNK signaling. These findings suggest that FST is a novel chemoattractant for migration and invasion of placental trophoblasts of mice.

Normal development and function of the placenta is essential for successful pregnancy. The major cell type of the placenta responsible for its essential functions is the trophoblast, which can invade the decidualized maternal endometrium, remodel the spiral arteries, and develop a nutrient-supplying function placenta in early pregnancy. During placental development in the mouse, blastocysts implant into the uterus on embryonic day 4.5 (E4.5), and meanwhile it is triggered by the surrounding maternal endometrial stromal cells to undergo a significant reaction termed decidualization [33]. At E8.5, fusion of the chorion and allantois marks the beginning of forming a mature placenta, in which a process such as abnormal fusion is the most common cause of embryonic lethality in the second trimester (E 9.5–14.5) [34]. Therefore, the placenta of E8.5 mice was chosen in this study to isolate placental trophoblasts because it is a time of active invasion in early pregnancy and because separation of the fetus, fetal membranes, and decidual tissues from the surface of the placenta are relatively easy to accomplish.

FST, as a secreted gonadal protein and activin-binding protein, is expressed in almost all tissues, and linked to female reproduction, with concentration elevation in plasma in the early stage of normal pregnancy [35,36,37]. Mice with uterine FST knockdown have severe infertility caused by failure to decidualize to support an embryo implantation [12]. FST has two major isoforms, FST288, which is anchored to the cell surface by interactions with heparin sulfate proteoglycans, and FST315, which is the predominant form in circulation. FST isoforms are implicated in uterine development that FST288 isoform is sufficient for the development, and that loss of FST315 isoform results in fertility defects that resemble activin hyperactivity and premature ovarian failure [38]. It was reported that FST enhanced human umbilical vein endothelial cell (HUVEC) migration, vascularization, and osteogenesis contributing [39]. In addition, TGF-β1-3, activin subunits, FST and follistatin-like 3 (FSTL3) were expressed in the placenta [40]. Interestingly, FST likely promoted trophoblast differentiation into giant cells, which could be our new research in the future [41]. In this study, we found that FST induced migration and invasion of primary cultured trophoblasts of mice in a time-dependent and dose-dependent manner in vitro, indicating that it has important implications for placental function in pregnancy.

The transformation of epithelial cells into motile mesenchymal cells, a process known as EMT, is integral in development and wound healing and contributes pathologically to fibrosis and cancer progression [42,43,44]. During the implantation of embryos, trophoblasts had some tumor behavior, such as EMT and actin cytoskeleton reorganization to enable dynamic cell elongation and directional motility [15,45]. Ezrin, a membrane-cytoskeleton junction protein, plays an essential role in cell polarity establishment, cell migration, and invasion [31]. In this study, we found that FST elevated expression of ezrin protein and the localization of ezrin was concentrated in the anterior margin of primary cultured trophoblast of mice towards FST gradient. It is essential to reorganize the cytoskeletal architecture and polarity complexes in EMT, which results in cell morphology changes such as cell elongation, membrane protrusions, and front-rear polarity, which in turn enables directional migration. During EMT, cells express low levels of epithelial proteins, for example, E-Cadherin, to reinforce the destabilization of adheres junctions, and acquire an affinity for mesenchymal cells through N-Cadherin interactions [46,47]. Moreover, remodeling of the extracellular matrix (ECM) during EMT is correlated with the increased expression of proteases, such as the matrix metalloproteinases 2 (MMP2) that enhances ECM protein degradation to facilitate invasion [48]. Our study revealed that FST might promote migration and invasion of primary cultured trophoblasts of mice by regulating the expression of N-Cadherin, E-Cadherin, vimentin, and MMP2 to achieve EMT and ECM degradation. Calcium signaling is important for cell migration and cell-substrate adhesion and contraction of the actomyosin network [49]. In the present study, we found that FST increased intracellular calcium levels of primary cultured trophoblasts of mice. The above results suggest that migration and invasion of trophoblasts induced by FST are associated with EMT and ECM degradation through calcium signaling.

Activin A induces invasion of human trophoblasts by up-regulating N-Cadherin expression in a Smad3-dependent manner [16], and increases snail-mediated MMP2 expression through ALK4 [50]. Mitogen-activated protein kinases (MAPK) signaling pathway involves p38 mitogen-activated protein kinases (p38-MARK) [51], extracellular signal-regulated kinases 1 and 2 (ERK1/2) [52,53], and c-Jun NH2-terminal kinase (JNK) in a cell type-specific manner [54]. These signaling proteins can transduce extracellular signals from the activin A ligand to the nucleus to cause the corresponding cell biological effects such as cell proliferation, apoptosis, differentiation, and migration. Previous studies have reported that MAPKs activation plays an important role in the process of EMT [55,56]. Cell proliferation and production of ECM proteoglycans and collagen II expression were decreased in follistatin-like protein 1 (FSTL1)-deficient mesenchymal stem cells (MSCs) and TGF-β signaling including Smad3, p38 MAPK, and Akt was significantly suppressed [57]. FSTL1 induced MMP3 and MMP13 gene expression in rheumatoid arthritis synoviocytes requiring MAPK, JAK/STAT3 and NF-κB pathways [58]. In line with ERK1/2, JNK seems to be essential for phosphorylation of activating protein 1 (AP1), which is caused by stress and exposure to various cytokines [59]. It was reported that osteosarcoma cell metastasis was suppressed in vivo via blocking the p38 and JNK-mediated cAMP response element-binding protein (CREB)-activating protein 1 (AP1) complexes formation [60]. Trophoblast invasion was also mediated by the AP1 transcription factors c-FOS and c-JUN [61]. Restrained MAPK-ERK and AP-1 signaling resulted in poor migration and invasion of trophoblast cell line HTR-8/SVneo cells [62]. Our present study revealed that FST induced migration and invasion of mice trophoblasts of up-regulating EMT-related proteins by JNK signaling. The effects were weakened by the treatment of the JNK inhibitor. In neuroendocrine cells, JNK plays an essential role in nerve growth factor (NGF) enhancement of Ca^2+^-evoked release [63]. It has also been reported that the activation of the JNK pathway could induce the increasing of mitochondrial Ca^2+^ influx in MCF-7 and MCF-7/MDR cells [64]. Our data indicated that FST-induced migration and invasion of trophoblasts was also related to increase in intracellular calcium flux and enhancing the activation of JNK signaling.

## 5. Conclusions

In summary, our findings advance the novel concept that FST is a novel chemoattractant for the migration of placental trophoblasts of mice in vitro. These data suggest that FST can induce migration and invasion of primary cultured trophoblasts of mice due to EMT and ECM degradation by enhancing JNK signaling. Thus, this study sheds a new light on understanding the role of FST in regulation of placental trophoblast activities.

## Figures and Tables

**Figure 1 cells-11-03816-f001:**
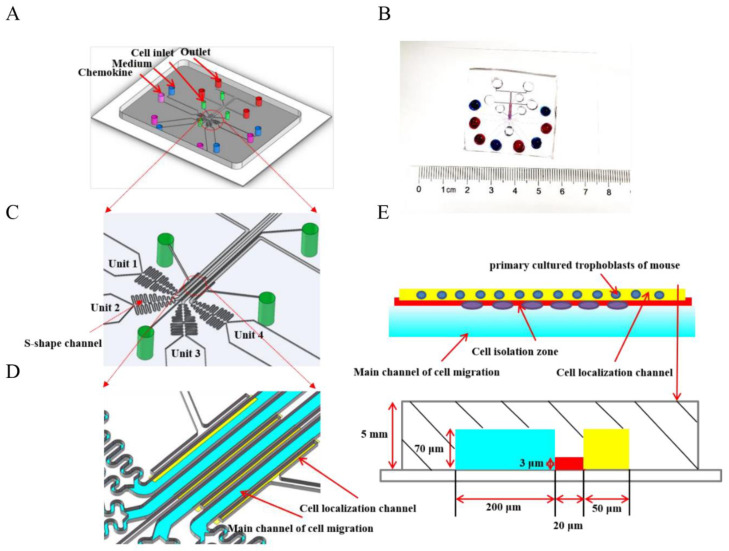
Illustration of the microfluidic D^4^-Chip device. (**A**,**C**) A schematic illustration of the fourth docking microfluidic device containing four units. Every unit is made of a cell inlet, outlet, and a dosing inlet containing medium and chemokine. (**B**) Soft-lithography adheres to the glass plate, and the dosing inlets are added to the ink. Blue ink represents medium and red ink represents chemokine. Moreover, both of them form the concentration gradient of chemokine in the main channel of cell migration. (**D**) From the image of the four microfluidic channels, cells are aligned beside in cell localization channel, and the formation of a concentration gradient in the main channel of cell migration owes to adding medium and chemokine at the same time. (**E**) I The length of the main channel of cell migration (Brillant blue), cell isolation zone (Red), and cell localization channel (Yellow) are 200 μm, 20 μm and 50 μm, respectively. What is more, illustration of the microfluidic migration assay simply involves cell seeding and solution loading followed by incubation and final end-point imaging analysis.

**Figure 2 cells-11-03816-f002:**
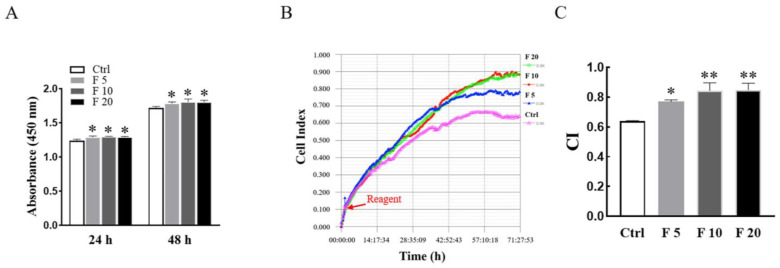
Effects of FST on viability and proliferation of primary cultured trophoblasts of mice. (**A**) The primary cultured trophoblasts of E8.5 mice were treated with FST for 24 h and 48 h, respectively, and the viabilities of cells were examined by the CCK-8 assay. The graph shows the absorbance of OD450 (*n* = 6). * *p* < 0.05, compared with control. (**B**) The proliferation of primary cultured trophoblasts of mice treated with FST for 72 h was examined by RTCA. The red arrow represents the dosing point. Ctrl: 2% FBS culture medium control. F5: 5 ng/mL of FST. F10: 10 ng/mL of FST. F20: 20 ng/mL of FST. (**C**) The graph showed the average cell index and standard deviation (*n* = 3). * *p* < 0.05, ** *p* < 0.01 compared with control.

**Figure 3 cells-11-03816-f003:**
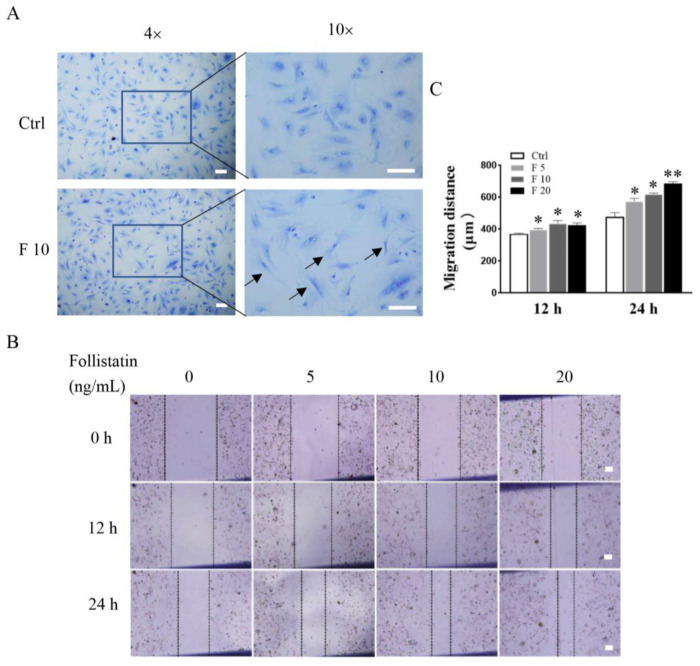
Effects of FST on morphology and wound healing of primary cultured trophoblasts of mice. (**A**) Morphology of mice trophoblasts were determined by Giemsa staining. Arrows indicated the typical morphological changes of cells treated with 10 ng/mL FST for 24 h. (**B**) The scratch-wound model was generated in monolayer trophoblasts, and then cells were treated with FST for 12 h and 24 h, respectively. (**C**) The graph showed the degree of wound healing (*n* = 3). Scale bar = 100 μm. Ctrl: 2% FBS culture medium control. F5: 5 ng/mL of FST. F10: 10 ng/mL of FST. F20: 20 ng/mL of FST. * *p* < 0.05. ** *p* < 0.01, compared with control.

**Figure 4 cells-11-03816-f004:**
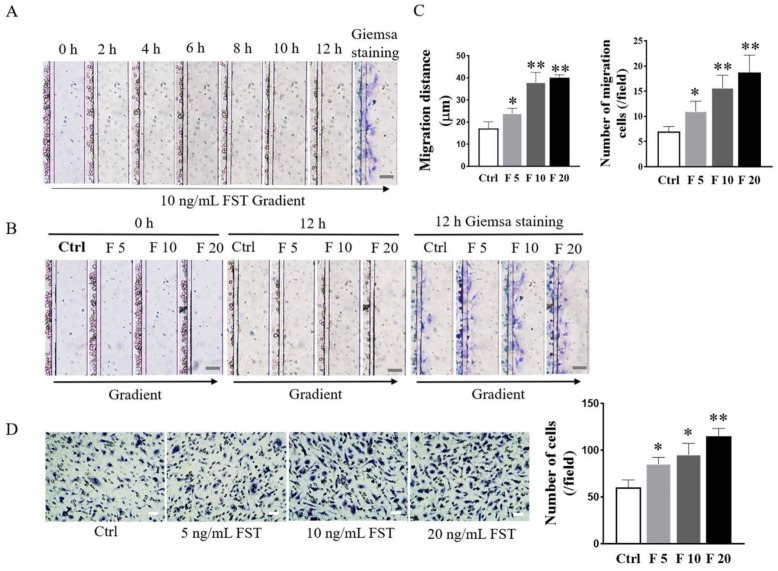
Effects of FST on migration and invasion of mice trophoblasts. (**A**) Images of trophoblast migration towards 10 ng/mL FST gradient were taken every 2 h in microfluidic device. (**B**) Images of trophoblast migration towards 0, 5, 10, and 20 ng/mL FST gradients were taken in the microfluidic device at 0 h and 12 h, respectively. Scale bar = 100 μm. (**C**) The graph showed the average migrated distance (**left**) and the number of migrated cells (**right**) in the same size fields of microfluidic device (*n* = 3). Ctrl: 2% FBS culture medium control. F5: 5 ng/mL of FST. F10: 10 ng/mL of FST. F20: 20 ng/mL of FST. * *p* < 0.05, ** *p* < 0.01, compared with control. (**D**) Invasion of trophoblasts treated with FST was determined by the Matrigel-coated transwell chamber invasion assay. Representative images of cell invasion were shown and the graph represented the average number of stained cells counted in five randomly chosen fields from each chamber (*n* = 3). Scale bar = 50 μm. * *p* < 0.05. ** *p* < 0.01, compared with control.

**Figure 5 cells-11-03816-f005:**
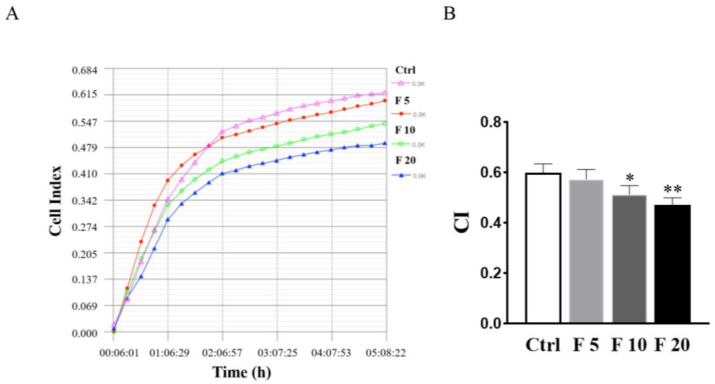
Effects of FST on adhesion of primary cultured trophoblasts of mice. (**A**) The cell adhesion was examined by RTCA in primary cultured mice trophoblasts subject to FST for 5 h. (**B**) The graph showed the average cell index (CI) and standard deviation (*n* = 3). Ctrl: 2% FBS culture medium control. F5: 5 ng/mL of FST. F10: 10 ng/mL of FST. F20: 20 ng/mL of FST. * *p* < 0.05, ** *p* < 0.01 compared with control.

**Figure 6 cells-11-03816-f006:**
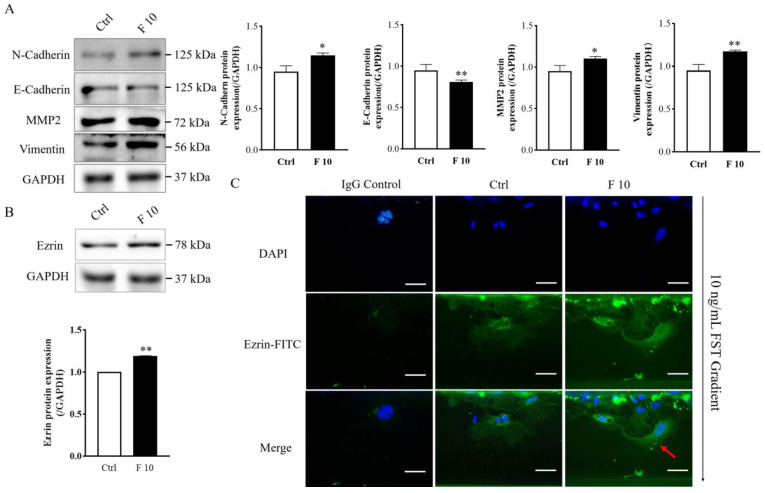
Effects of FST on expression of migration-related protein and polarization of mice trophoblasts. (**A**) Migration-related protein expression was examined by Western blotting in primary cultured trophoblasts treated with 2% FBS culture medium (Ctrl) and 10 ng/mL FST (F10) for 24 h. The graph represents the relative levels of protein expression (*n* = 3). The expression levels of protein were normalized against GAPDH expression. (**B**) Ezrin expression in primary cultured trophoblasts of mice were analyzed by Western blotting. The graph represented the relative levels of ezrin protein expression (*n* = 3). (**C**) Ezrin distribution in primary cultured trophoblasts of mice were examined by immunofluorescent staining in microfluidic chip. IgG control, trophoblasts were stained with the normal rabbit IgG as isotype IgG control taking the place of anti-Ezrin antibody. Ctrl: trophoblasts that were treated with 2% FBS culture medium were stained with anti-Ezrin antibody. F 10, trophoblasts that were treated with 10 ng/mL FST were stained with anti-Ezrin antibody. Red arrows indicated the typical changes of Ezrin distribution in cells treated by 10 ng/mL FST. Scale bar = 50 μm. * *p* < 0.05. ** *p* < 0.01, compared with control.

**Figure 7 cells-11-03816-f007:**
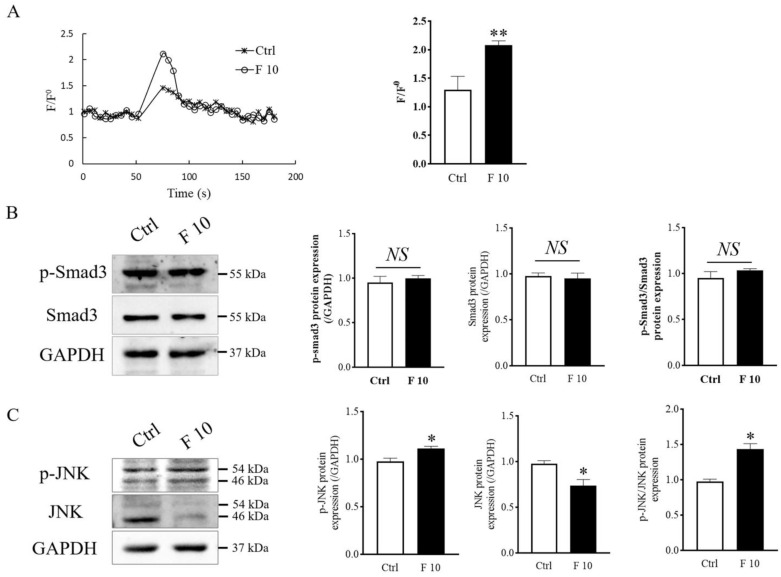
Effects of FST on calcium signaling, Smad3, and JNK signaling of mice trophoblasts. (**A**) The calcium levels were measured by the Fluo-4 signal intensity normalized to the baseline (F/F^0^). The graph represented the comparison of the peak value of calcium signal upon stimulation under 2% FBS culture medium (Ctrl) and 10 ng/mL FST (F10) (*n* = 3). ** *p* < 0.01, compared with control. (**B**,**C**) Levels of Smad3, p-Smad3, JNK, and p-JNK proteins were examined by Western blotting in mice primary trophoblasts subjecting to 10 ng/mL FST for 4 h. The graph represented the relative level of proteins (*n* = 3). The levels of protein were normalized against GAPDH expression. * *p* < 0.05. ** *p* < 0.01, compared with control.

**Figure 8 cells-11-03816-f008:**
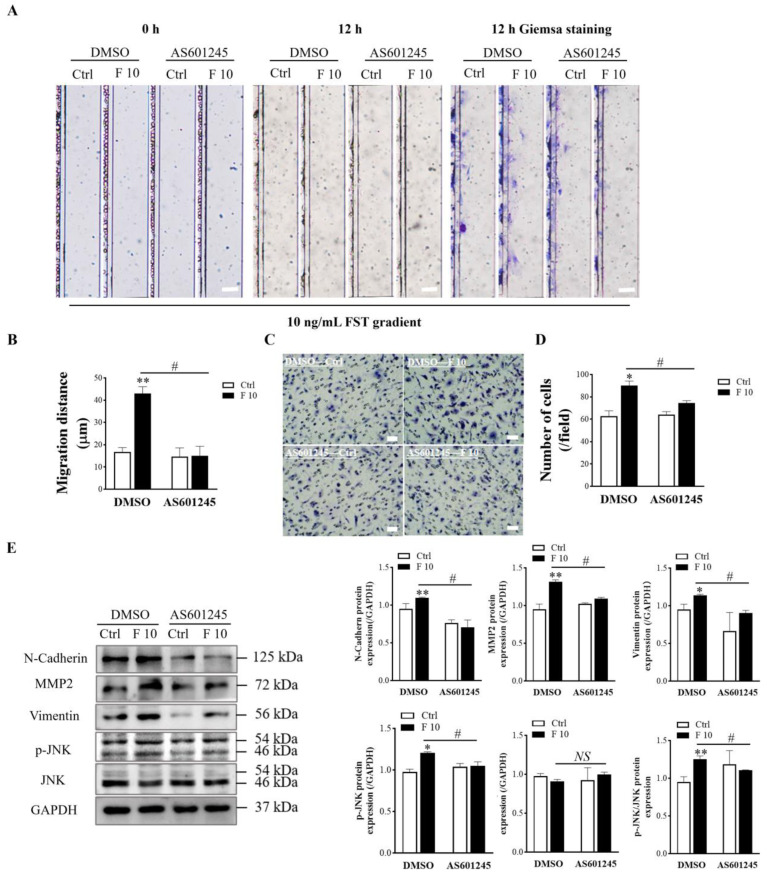
Effects of JNK inhibitor AS601245 on FST-induced migration and invasion of primary cultured trophoblasts of mice. (**A**) The primary cultured trophoblasts were pre-treated for 1 h with vehicle control (DMSO) or 1 μmol/L AS601245, and then migration of trophoblasts was determined by D^4^ chip for 12 h. Scale bar = 100 μm. (**B**) The graph showed the average migration distance of cells in the same size fields of microfluidic device (*n* = 3). (**C**) The invasion of primary cultured trophoblasts pre-treated with vehicle control (DMSO) or 1 μmol/L AS601245 was determined by Matrigel-coated transwell chamber invasion assay for 12 h. Scale bar = 50 μm. (**D**) The graph represents the average number of stained cells counted in five randomly chosen fields from each chamber (*n* = 3). (**E**) Levels of N-Cadherin, MMP2, vimentin, p-JNK, and JNK proteins in trophoblasts pre-treated with vehicle control (DMSO) or 1 μmol/L AS601245 in DMSO were determined by Western blotting with GAPDH as the internal control protein. The graph represents the relative level of proteins (*n* = 3). Ctrl: 2% FBS culture medium control. F10: 10 ng/mL of FST. * *p* < 0.05, ** *p* < 0.01, compared with vehicle control. # *p* < 0.05. compared with the indicted DMSO group.

## Data Availability

The data are contained within the article and Appendix A.

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
