# Peer review of "Follistatin Is a Novel Chemoattractant for Migration and Invasion of Placental Trophoblasts of Mice"

_cells, 2022, doi:10.3390/cells11233816_

Round 1

Reviewer 1 Report

This manuscript provides some novel data on the role of Follistatin on trophoblast invasion, and the authors are also able to tease out some of the signaling pathways involved in this effect. Overall, the data are robust. However, several aspects need to be addressed to improve the accurancy and impact of this manuscript.

1. The authors describe the process of trophoblast invasion, this is very different from "embryo implantation". The authors shoudl remove embryo implantation throughout and rephrase these sentences to more accurately reflect the processes that their study addresses.

2. Do the authors use the undigested tissue clumps for their analyses, or the single cells? This is not clear in the Materials and Methods. Pls see also below.

3. Are the invading cells indeed trophoblast cells? This should be verified by staining of an invasion filter and/or by staining of the migratory cells in the wound healing assay.

4. The authors should also take into consideration the data from Natale et al, Dev. Biol 2009. This paper shows that Follistatin likely promotes trophoblast differentiation into giant cells. That would explain the authors's results. The authors should specifically test for this in their primary cell culture system, for example by using a panel of gene markers in RT-PCRs of cells with and without FST addition.

5. The assertion of Ezrin at the leading edge is entirely unclear. No negative cells can be seen in the Figure. Please provide an image that shows these data more clearly.

6. In Figure 8, it is unclear what statistical comparisons have been performed. What was the statistical method applied to these data? T-tests would not be approproiate.

7. Please detail in the figure legends the exact number of experiments that were analyzed for the generation of every bar graph, throughout the manuscript. The simple assertion "experiment has been performed more than three times" is insufficient.

8. Fig 7c, there is no total JNK signal at all in the FST10 condition. Also, the total JNK and p-JNK blots look entirely different between Figures 7c and 8e. Please provide better and more consistent examples of blots. Please also label the size of bands according tothe standard marker that was presumably run on every gel [kDa].

9. The manuscript requires some language editing for English throughout.

Author Response

Dear Reviewer 1:

Thank you for your comments on manuscript entitled “Follistatin is a Novel Chemoattractant for Migration and Invasion of Placental Trophoblasts of Mouse” (ID: cells-1949370). These comments are all valuable and helpful for improving our article. The manuscript has been revised carefully according to your comments, and the following are our responses to your comments.

  1. The authors describe the process of trophoblasts invasion, this is very different from "embryo implantation". The authors should remove embryo implantation throughout and rephrase these sentences to more accurately reflect the processes that their study addresses.

Response: Thank you for your suggestions. It is really true as Reviewer suggested that trophoblast invasion is not equal to embryo implantation. The process of embryo implantation will be accompanied by the invasion and migration of trophoblasts into the decidua of the uterus for uterine spiral arteries remodeling and formation of a nutrient-supplying functional placenta. So, we have revised the manuscript according to Reviewer’s comments.

  1. Do the authors use the undigested tissue clumps for their analyses, or the single cells? This is not clear in the Materials and Methods.

Response: Thank you for your comments. We are very sorry for unclear writing in Materials and Methods. In this study, the digested single cells of placental tissues were used. The methods of manuscript were revised carefully.

  1. Are the invading cells indeed trophoblast cells? This should be verified by staining of an invasion filter and/or by staining of the migratory cells in the wound healing assay.

Response: Thank you for your suggestions. In this study, mouse placentas at embryonic day 8.5 (E8.5) were culled and the digested single cells of placental tissues were stained by immunofluorescence staining with anti-cytokeratin-7 (CK7) and Vimentin antibodies (Figure S1 in Supplementary Figure). Cytokeratin-7 (CK7), as trophoblasts marker, was expressed positively in the cells, and vimentin, as marker of mesenchymal cells, was expressed negatively. So, primary cultured placental cells belong to trophoblasts. The description of the CK7 staining results were added in the Results 3.1 of revised manuscript (Figure S1 in Supplementary Figure). Additionally, the migratory cells in the wound healing assay were also stained with cytokeratin-7 (CK7) (Figure S2 in Supplementary Figure), which confirmed that the migratory cells in the wound healing assay were indeed trophoblast cells stained for CK7 positively.

  1. The authors should also take into consideration the data from Natale et al, Dev. Biol 2009. This paper shows that Follistatin likely promotes trophoblast differentiation into giant cells. That would explain the authors' results. The authors should specifically test for this in their primary cell culture system, for example by using a panel of gene markers in RT-PCRs of cells with and without FST addition.

Response: Thank you for your suggestions. This is a perfect assumption that Follistatin likely promotes trophoblast differentiation into giant cells. We added this paper (Dev. Biol 2009) into References, and it would be interesting to investigate this question in future studies.

  1. The assertion of Ezrin at the leading edge is entirely unclear. No negative cells can be seen in the Figure. Please provide an image that shows these data more clearly.

Response: Thank you for your suggestions. We have added the image of negative cells in the revised manuscript and revised the Figure 6 according to Reviewer’s comments.

  1. In Figure 8, it is unclear what statistical comparisons have been performed. What was the statistical method applied to these data? T-tests would not be appropriate.

Response: Thank you for your suggestions. Student t-test is applicable to the comparison of two groups of data. One-way ANOVA followed by Tukey’s multiple comparisons test is applicable to the comparison of data of multiple groups in Figure 8 and one-way ANOVA test had been shown in statistical analysis of Materials and Methods 2.13.

  1. Please detail in the figure legends the exact number of experiments that were analyzed for the generation of every bar graph, throughout the manuscript. The simple assertion "experiment has been performed more than three times" is insufficient.

Response: Thank you for your suggestions. We have added the number of experiments (n = 3 or 6) in revised manuscript.

  1. Fig 7c, there is no total JNK signal at all in the FST10 condition. Also, the total JNK and p-JNK blots look entirely different between Figures 7c and 8e. Please provide better and more consistent examples of blots. Please also label the size of bands according to the standard marker that was presumably run on every gel [kDa].

Response: Thank you for your suggestions. We have provided better and more consistent blots and added the analysis for total JNK and p-JNK. And the size of bands was labelled according to the standard marker [kDa] in the revised manuscript.

  1. The manuscript requires some language editing for English throughout.

Response: Thank you for your suggestion. The language has been edited carefully by our collaborator Dr. Qi from University of Maryland, Baltimore, USA.

Once again, thank you for your good comments.

Sincerely yours,

Zhonghui Liu, Ph.D.,

Professor,

Department of Immunology, College of Basic Medical Sciences, Jilin University

126 Xinmin Street, Changchun 130021, China

Tel.: +86 431 8561 9476

E-mails: liuzh@jlu.edu.cn

Reviewer 2 Report

This manuscript investigates the potential role of follistatin (FST) to function as a chemoatractant in the viability, migration, adhesion, proliferation and invasiveness of primary cultured placental trophoblasts from embryonic day 8.5 mice in conventional assays and using novel microfluidic device used to monitor cell migration in the absence of FST and gradients of 5, 10, and 20 ng/ml FST. Treatment with FST reduced adhesion of trophoblasts and promoted upregulation of EMT and increased intracellular calcium transients. Western blot analysis demonstrated elevation of p-JNK and the ratio of p-JKN/JNK and expression of proteins associated with EMT which were suppressed by a specific JNK inhibitor.  The authors conclude that elevated FST during pregnancy may act as a cytokine capable of supporting trophoblast function by migration and invasion through enhanced JNK signaling to support embryo implantation.

This is an interesting and informative report that appears to be appropriately designed and comprehensive.  Overall, the findings are of considerable interest, however, there is a need for considerable English editing that would tighten up the presentation of the findings, several of which are listed below. Use of spell checker is needed.

Line 62-63.  Change …trophoblasts transform from an epithelial to a mesenchymal phenotype…

Line 97. The meaning of this sentence is unclear.  Were undigested tissues used to harvest additional trophoblasts?

Line 98-100.  Sentence not clear- do the authors mean:   From mouse placentas collected on E8.5, approximately 40,000 viable trophoblast cells were recovered with at least 85% of the cells testing positive for expression of cytokeratin-7 as a trophoblast marker.

Line 105.  It is not clear what “et al” refers to.

Line 126. The relevance or significance of the cell index (CI) should be specified.   Electrode impedance, which is displayed and recorded as Cell Index (CI) values, reflect the biolgical status of monitored cells, including the cell number, viability, morphology and adhesion.

Line 176.  Delete “artificially”

Line 178 .  Should “distance be replaced with “migration”

Line 176.  What is the meaning of “artificially measured”

Line 178.  Should  “average cell distance” be replaced with average cell migration distance?

Line 219  the NA of the 20x lens should be specified.

Line 227. Calcium flux assay:  Fluo-4 fluorescence intensity

Line 268. Meaning of “…FST were long and multiplied by giemsa staining …”  is unclear

Figure 2.  There is no (C) for the migration distance in the legend.

Line 292. Replace “more increasing invasion…” with “increased invasion of primary mouse trophoblasts…”

Figure 4.  Correct Labeling of Figure 4 C and D.

Line 313. Delete “potential”

Figure 5.  Figure legends should refer to graphs A and B

Line 349.  Replace “flow” with flux or transient

Figure 6. Immuofluorescence control is not described.  It appears theat omission of primary antibody was meant to be a control but this is only a control for the specificity of the secondary antibody.

Figure 7A.  A single for 10 nm/ml is shown representing three separate experiments.  Methods reports o, 10, and 20 nm/ml doses.  Was there a dose response?   Should a standard error bar be included? 

Line 407.  Meaning of sentence is unclear

Line 414 – 416.  Sentence unclear.  Perhaps:  Therefore, the placenta of E8.5 mouse was chosen in this study to isolate placental trophoblasts because it is a time of active invasion in early pregnancy and because separation of the fetus, fetal membranes and decidual tissues from the surface of the placenta is relatively easy to accomplish

Line 420. Replace “fail” with failure

435.  Is heteroplastid embryos the proper terminology here?

Author Response

Dear Reviewer 2:

Thank you for your comments on manuscript entitled “Follistatin is a Novel Chemoattractant for Migration and Invasion of Placental Trophoblasts of Mouse” (ID: cells-1949370). These comments are all valuable and helpful for improving our article. The manuscript has been revised carefully according to your comments, and the language has been edited by our collaborator Dr. Qi from University of Maryland, Baltimore, USA. The following are our responses to your comments.

1.Line 62-63. Change …trophoblasts transform from an epithelial to a mesenchymal phenotype…

Response: Thank you for your suggestion. We corrected the statements.

  1. Line 97. The meaning of this sentence is unclear. Were undigested tissues used to harvest additional trophoblasts?

Response: Thank you for your suggestion. We are very sorry for unclear writing about experimental methods. We have corrected this sentence. In this study, the tissues were digested for harvesting single trophoblasts. Briefly, Chopped tissues were added to serum-free medium containing 2 mg/mL DNase and 0.1 mg/mL liberase, and digested at 37 °C for 40 min. Then, the undigested tissues were separated using a cell strainer, and the obtained suspension containing digested single placental trophoblasts were centrifuged 300 g for 5 min for next experiments.

  1. Line 98-100. Sentence not clear- do the authors mean: From mouse placentas collected on E8.5, approximately 40,000 viable trophoblast cells were recovered with at least 85% of the cells testing positive for expression of cytokeratin-7 as a trophoblast marker.

Response: Thank you for Reviewer’s suggestions. We have corrected these sentences. The sentences mean that more than 40,000 living trophoblast cells were obtained from a mouse placenta, and at least 95% of the living cells expressed CK-7.

  1. Line 105. It is not clear what “et al” refers to.

Response: Thank you for Reviewer’s suggestion. We have deleted ‘et al’ in revised manuscript.

  1. Line 126. The relevance or significance of the cell index (CI) should be specified. Electrode impedance, which is displayed and recorded as Cell Index (CI) values, reflect the biological status of monitored cells, including the cell number, viability, morphology and adhesion.

Response: Thank you for Reviewer‘s good comments. We have corrected this sentence according to Reviewer‘s comments.

6.Line 176. Delete “artificially”, line 176.  What is the meaning of “artificially measured”

Response: Thank you for Reviewer‘s suggestion. We have deleted “artificially”. The sentence means we want to expressed that average cell movement trajectories was manually tracked in ImageJ.

  1. Line 178. Should “distance” be replaced with “migration”. Line 178. Should “average cell distance” be replaced with average cell migration distance?

Response: Thank you for Reviewer’s good suggestion. We corrected this sentence with “average cell migration distance”.

  1. Line 219 the NA of the 20x lens should be specified.

Response: Thank you for Reviewer’s comments. The statement was corrected with the Zeiss 20x (0.8NA) Plan -Apochromat objective lens. A Zeiss Axio Imager Z1 microscope equipped with an Apotome device (Carl Zeiss) was used to capture images using a Zeiss Plan APOCHROMAT 20x/0.8na.

  1. Line 227. Calcium flux assay: Fluo-4 fluorescence intensity

Response: Thank you for Reviewer’s suggestion. “Calcium flux assay” were corrected with “Calcium flux assay: Fluo-4 fluorescence intensity”.

  1. Line 268. Meaning of “…FST were long and multiplied by giemsa staining …” is unclear

Response: Thank you to Reviewer for good comments. We have corrected the statement to that the morphology of trophoblasts treated with 10 ng/mL of FST exhibited longer and polygonal shape compared with trophoblasts in control group.

  1. Figure 3. There is no (C) for the migration distance in the legend.

Response: Thank you for Reviewer’s suggestion. We have added (C) for the migration distance in the legend.

  1. Line 292. Replace “more increasing invasion…” with “increased invasion of primary mouse trophoblasts…”

Response: Thank you for Reviewer’s suggestion. “more increasing invasion…” were corrected with increased invasion of primary mouse trophoblasts…”.

  1. Figure 4. Correct Labeling of Figure 4 C and D.

Response: Thank you for Reviewer’s comments. Labeling of Figure 4 C and D was corrected.

  1. Line 313. Delete “potential”

Response: Thank you for Reviewer’s suggestion. “potential” was deleted.

  1. Figure 5. Figure legends should refer to graphs A and B

Response: Thank you for Reviewer’s comments. The A and B have added to figure legends.

  1. Line 349. Replace “flow” with flux or transient

Response: Thank you for Reviewer’s comments. “Flow” was corrected with “flux”.

  1. Figure 6. Immunofluorescence control is not described.  It appears theat omission of primary antibody was meant to be a control but this is only a control for the specificity of the secondary antibody.

Response: Thank you for Reviewer’s comments. The immunofluorescence isotype IgG control with normal rabbit IgG was added in revised manuscript.

  1. Figure 7A. A single for 10 nm/ml is shown representing three separate experiments. Methods reports 0, 10, and 20 nm/ml doses. Was there a dose response?   Should a standard error bar be included? 

Response: Thank you for Reviewer’s comments. We have corrected the Methods with 0 and 10 ng/mL FST. The graph represented the standard error of the peak value (n = 3) in Figure 7A.

  1. Line 407. Meaning of sentence is unclear

Response: Thank you for Reviewer’s suggestion. The sentence was corrected as “Normal development and function of the placenta is essential for successful pregnancy. The major cell type of the placenta responsible for its essential functions is the trophoblast, which can invade the decidualized maternal endometrium, remodel the spiral arteries and develop a nutrient-supplying function placenta in early pregnancy.”.

  1. Line 414 – 416. Sentence unclear. Perhaps:  Therefore, the placenta of E8.5 mouse was chosen in this study to isolate placental trophoblasts because it is a time of active invasion in early pregnancy and because separation of the fetus, fetal membranes and decidual tissues from the surface of the placenta is relatively easy to accomplish

Response: Thank you for Reviewer’s good suggestion. The sentence was corrected as “because separation of the fetus, fetal membranes and decidual tissues from the surface of the placenta is relatively easy to accomplish”.

  1. Line 420. Replace “fail” with failure

Response: Thank you for Reviewer’s suggestion. “Fail” was corrected with “failure”.

  1. 435. Is heteroplastid embryos the proper terminology here?

Response: Thank you for Reviewer’s suggestion. We have corrected the sentence.

Once again, thank you for your good comments.

Sincerely yours,

Zhonghui Liu, Ph.D.,

Professor,

Department of Immunology, College of Basic Medical Sciences, Jilin University

126 Xinmin Street, Changchun 130021, China

Tel.: +86 431 8561 9476

E-mails: liuzh@jlu.edu.cn

Reviewer 3 Report

The manuscript: “Follistatin is a novel chemoattractant for migration and invasion of placental trophoblasts of mouse.” Describe that mouse treated with FST showed increased trophoblast migration indicating FST being a pregnancy chemoattractant. The authors show that FST impacts migration and invasion of placental trophoblasts in mouse experiments. The manuscript contains a clear introduction and methods are described properly. Results and figures are presented clear and especially the table of content attracts my attention. For me the manuscript is written comprehendible and therefore should be ready for publication.

Author Response

Dear Reviewer 3:

Thank you for your comments on manuscript entitled “Follistatin is a Novel Chemoattractant for Migration and Invasion of Placental Trophoblasts of Mouse” (ID: cells-1949370). We will continue to work hard in future. Additionally, The revised manuscript was resubmitted according to Editor’s letter.

Sincerely yours,

Zhonghui Liu, Ph.D.,

Professor,

Department of Immunology, College of Basic Medical Sciences, Jilin University

126 Xinmin Street, Changchun 130021, China

Tel.: +86 431 8561 9476

E-mails: liuzh@jlu.edu.cn

Round 2

Reviewer 1 Report

The authors have done a thorough job with these revisions.

I cannot see all the figures in the revised document. Nevertheless, trusting the rebuttal letter the revisions are adequate.

Minor point: please clarify/rephrase this sentence in the Abstract:

"Migration and invasion of placental trophoblasts towards the decidualized uterine tissue are in favor of remodeling of uterine spiral arteries remodeling and placental development." This sentence does not make sense.

Author Response

Dear Reviewer:

Thank you for your comments on the manuscript entitled “Follistatin is a Novel Chemoattractant for Migration and Invasion of Placental Trophoblasts of Mouse” (ID: cells-1949370). These comments are all valuable and helpful for improving our article. The manuscript has been revised carefully according to your comments. We appreciate for your careful reading of our work, and hope that the correction will meet with approval. The following are our responses to your comments.

1 “I cannot see all the figures in the revised document. Nevertheless, trusting the rebuttal letter the revisions are adequate.”

Response: Thank you for your comments. We apologize for this error which did not appear all figures because that during the conversion from marked Microsoft Word doc document to PDF document “figure images” disappeared. All the figures can be seen in the new revised manuscript.

2 Minor point: please clarify/rephrase this sentence in the Abstract: "Migration and invasion of placental trophoblasts towards the decidualized uterine tissue are in favor of remodeling of uterine spiral arteries remodeling and placental development." This sentence does not make sense.

Response: Thank you for your good suggestions. We have corrected the statement with “Trophoblasts migration and invasion into the endometrium are critical events in the placental development”.

Once again, thank you for your good comments.

Sincerely yours,

Zhonghui Liu, Ph.D.,

Professor,

Department of Immunology, College of Basic Medical Sciences, Jilin University

126 Xinmin Street, Changchun 130021, China

Tel.: +86 431 8561 9476

E-mails: liuzh@jlu.edu.cn